# An Underwater Acoustic Network Positioning Method Based on Spatial-Temporal Self-Calibration

**DOI:** 10.3390/s22155571

**Published:** 2022-07-26

**Authors:** Chao Wang, Pengyu Du, Zhenduo Wang, Zhongkang Wang

**Affiliations:** National Key Laboratory of Science and Technology on Sonar, Hangzhou Applied Acoustics Research Institute, Hangzhou 310000, China; heaven663@163.com (P.D.); wangzhenduo890109@gmail.com (Z.W.); sklwzk@aliyun.com (Z.W.)

**Keywords:** beacon node drift, spatial-temporal self-calibration, networked positioning, underwater acoustic networks

## Abstract

The emergence of underwater acoustic networks has greatly improved the potential capabilities of marine environment detection. In underwater acoustic network applications, node location is a basic and important task, and node location information is the guarantee for the completion of various underwater tasks. Most of the current underwater positioning models do not consider the influence of the uneven underwater medium or the uncertainty of the position of the network beacon modem, which will reduce the accuracy of the positioning results. This paper proposes an underwater acoustic network positioning method based on spatial-temporal self-calibration. This method can automatically calibrate the space position of the beacon modem using only the GPS position and depth sensor information obtained in real-time. Under the asynchronous system, the influence of the inhomogeneity of the underwater medium is analyzed, and the unscented Kalman algorithm is used to estimate the position of underwater mobile nodes. Finally, the effectiveness of this method is verified by simulation and sea trials.

## 1. Introduction

The application of underwater acoustic networks (UANS) is becoming increasingly more extensive. Nodes in the network can better help scientists understand the underwater environment by collecting water parameters, such as temperature, salinity, pressure, etc.; commercial companies can use network mobile nodes to monitor and control submarine cables, pipelines, and other facilities. The underwater acoustic network has also been used in the military, such as monitoring submarines and ships in the network.

In various applications of underwater acoustic networks, node location is a basic and important task. Underwater acoustic network nodes, especially mobile nodes, such as autonomous underwater vehicles (AUVs), need node location information during data collection. For example, the data will be invalid if the location information is missing for the temperature data collected by mobile nodes. Because the electromagnetic waves have limited communication ranges due to the strong propagation in water, the widely used positioning system Globe Positioning System (GPS) is not feasible in the underwater environment [1,2].

People have developed many different positioning methods [2,3,4,5,6,7,8,9,10,11], taking advantage of the feature that acoustic signals can travel long distances in the underwater environment. The main idea of these methods is to obtain distance information from the reference node according to sound propagation characteristics, and then convert the distance information into the underwater node’s position information through signal processing. Distance information is usually characterized by the signal strength (SS) [3,4], angle of arrival (AOA) [5,6], time of arrival (TOA) [2], and time differences of arrival (TDOA) [2]. In the underwater acoustic environment, the strength of the received signal is not convenient since the propagation loss is difficult to obtain accurately in a time-varying environment. Using AOA for UANS has been considered in [7]; however, it has not been widely employed due to the size and cost of directional antennas. Distance measurement based on propagation time delay is widely used in underwater acoustic networks. This is because the speed of sound propagation in water is low (about 1500 m/s), and there is no need to receive signals with high time resolution. This paper is also based on the differences in signal arrival time for underwater acoustic network localization.

Distance-based location algorithms usually need to deal with non-linear filtering problems in the underwater acoustic environment. Most researchers [12,13] deal with this problem using the extended Kalman filtering (EKF) algorithm. However, the complex derivation of the Jacobian matrices in the EKF often leads to implementation difficulties. The unscented Kalman (UKF) [14,15,16,17] is an alternative to the EKF, which approximates the probability distribution of the state by a set of deterministically chosen sigma points and propagates the distribution through the non-linear equations. It does not need to calculate the Jacobian matrices and can obtain a better performance than the EKF. In this paper, when dealing with non-linear filtering problems, the UKF method is used uniformly to obtain high-precision results.

Long Baseline (LBL) positioning technology is a commonly used method for underwater acoustic positioning. It is necessary to place beacon nodes at designated locations. The node to be positioned can estimate its position by obtaining TOA or TDOA with the beacon nodes. However, in the process of beacon deployment, its position needs to be calibrated, which is very cumbersome in practical applications. The GPS Intelligent buoy (GIB) [8] system appeared to improve positioning efficiency. It contains a GPS receiver and a hydrophone. The real-time received GPS position is used as the position of the beacon node, and there is no need to calibrate the position of the beacon node. However, GIB is a centralized technique, and it does not provide location information for the target.

With the development of underwater acoustic communication technology in recent years, increasingly more researchers have paid attention to network positioning methods, which can realize the central and distributed positioning simultaneously by using buoys in UANS as beacon nodes. However, the acoustic modem of buoys is deployed at a certain depth and is softly connected to the buoy body through a cable in the process of positioning. Affected by the undercurrent, the modem drifts with the current, resulting in the horizontal position being inconsistent with the GPS position of the buoy body. If the GPS position of the buoy body is used as the position of the modem, it will cause the spatial position error of the beacon node. To reduce this impact of the current, Chen [18] adopted the analytic hierarchy process and grey correlation coefficient method to analyze the reliability of beacon nodes and selects nodes with a low drifting degree for the node location. However, the drift error was not eliminated, and the drift process still affected the positioning accuracy.

Most positioning methods [9,10,11] ignore the bending of sound rays and use a fixed sound speed to convert time into the distance, which leads to positioning errors in the underwater environment. Ramezani [19] proposed a moving target localization algorithm in an equal gradient sound velocity profile environment. It uses the nonlinear relationship between time delay and distance in an isogradient environment for positioning. However, the time–distance relationship is difficult to obtain in complicated marine environments or at large distances. To obtain the time–distance relationship in the complex environment, Zhu [12] took the sound speed as an unknown quantity and estimated it in real-time during positioning. However, this method is carried out under a synchronization system. The synchronization between the network node to be located and the beacon node is difficult to attain in the underwater environment. However, there are few studies on the sound ray correction problem based on ray theory in asynchronous systems. We need to find a positioning method suitable for the underwater environment in an asynchronous system. 

This paper proposes an underwater acoustic network positioning method based on spatial-temporal self-calibration in response to the above problems. This method can automatically calibrate the space position of the buoy modem using only the GPS position and depth sensor information obtained in real-time. Additionally, in an asynchronous system, the time delay difference is used as the observation to construct the measurement equation affected by sound ray bending, and the UKF method is used to estimate the position of the node to be located. This paper is arranged as follows. Section 2 introduces the principle and structure of the proposed system. Then, the positioning algorithm proposed by this paper and the posterior Cramér–Rao bound (PCRB) method is introduced. Finally, the proposed algorithm is validated by simulations and experiments.

## 2. Principle and Structure of the System

### 2.1. Structure of the Network Position System

Figure 1 is a schematic diagram showing the use of the networked positioning method to locate mobile nodes, with multiple buoy beacons on the sea surface (for example, there are four beacons in the diagram). The underwater mobile node to be located includes a depth sensor and an underwater acoustic modem. The beacon node includes a depth sensor, an underwater acoustic modem, and a GPS receiver. The beacon modem is connected with the anchor block on the seabed by a rope. The depth sensor is used to measure the depth of the modem, and the GPS device of the buoy can obtain the position and synchronize information. The modem is used to send and receive integrated communication and navigation signals, namely location signals, which can realize communication and navigation services at the same time. There is an asynchronous relationship between the underwater mobile node and four buoy beacons.

During the navigation of the mobile node, four buoy beacons synchronously send their specific location signals at different frequencies, and each location signal contains the depth and position information of the transmitting beacon modem. The total amount of information contained in the location signal is less than 15 bytes. The information is coded and modulated by the spread spectrum system, and the length of the location signal is less than 3 s.

The mobile node receives four location signals simultaneously in the form of frequency division multiple access (FDMA). The arrival time of different location signals is $t_{i},\;\;where\;i\in \left[1,4\right],$ and the mobile node uses the time differences of $t_{i}$ to estimate its position.

### 2.2. Working Principle of the System

A location method based on spatial-temporal self-calibration for UANS is proposed in this paper and the working principle of the system is shown in Figure 2. In the figure, the buoy handling process is on the left and the mobile node handling process is on the right. The dashed arrow represents transmission through the underwater acoustic channel. Firstly, the buoy receives the GPS position and depth sensor information in real-time. Due to the influence of ocean currents, there is a deviation between the GPS position and the modem position. The system uses the GPS position as observations and uses the modem position and velocity as the state variables to construct a drifting model of the buoy to calibrate the spatial position of the modem. The calibrated position information is sent to the mobile node through a location signal. Next, the positioned mobile node collects location signals periodically transmitted by multiple buoy nodes and estimates their arrival time. Since the mobile node receives the location signals of multiple buoy nodes at different locations, the arrival time error of positioning signals will become larger. The system performs the arrival time calibration processing to calibrate the arrival time of the received signal. Finally, the mobile node takes the arrival time difference of each beacon positioning signal as observation variables and its position as state variables, combined with the sound ray correction, and constructs a positioning model to calculate its position in real-time.

## 3. Principle of the Network Positioning Method Based on Spatial-Temporal Self-Calibration

This section introduces the realization principle of the algorithms, which is shown in Figure 2, including spatial position calibration of the beacon modem, arrival time calibration of the location signal, and high-precision positioning for the mobile node.

### 3.1. Spatial Position Calibration of the Beacon Modem

In the conventional method, the GPS position is directly used as the position of the beacon modem. As shown in Figure 1, the impact of ocean currents will cause deviations in the position of the beacon modem, which will eventually result in the positioning error of mobile nodes. Next, this paper reduces the position error of the buoy modem by spatial calibration.

The buoy modem is limited by the connecting rope, and its motion state is consistent with that of the buoy. We assume that the position of the buoy modem at time k is $u_{ik}={\left[ux_{ik},uy_{ik},uz_{ik}\right]}^{T}$ with velocity ${u^{\prime }}_{ik}={\left[uv_{ixk},uv_{iyk},uv_{izk}\right]}^{T}$. $u_{ik}$ and ${u^{\prime }}_{ik}$ are used as state variables and the constant velocity model as the beacon motion model to construct the motion equation of the beacon modem. The state model is given by:(1)$$\left[\begin{array}{l} u_{ik}\\ {u^{\prime }}_{ik} \end{array}\right]=F*\left[\begin{array}{l} u_{ik-1}\\ {u^{\prime }}_{ik} \end{array}\right]+G*w\left(k\right)$$
where w(k) represents acceleration disturbance noise. The matrix F relates the state of the previous time instant to the current one, and the matrix G represents the noise disturbance coefficient, which is shown in the following:(2)$$F=\left[\begin{array}{cc} I & T_{u}*I\\ 0 & I \end{array}\right]\;\mathrm{and}\;G=\left[\begin{array}{c} T_{u}{}^{2}/2*I\;\\ T_{u}*I \end{array}\right]$$
where $T_{u}$ is the update interval of state variables.

The buoy body will drift due to the influence of waves or currents. There is a soft connection between the modem and the buoy body, so the modem will drift along with the buoy body. However, their positional relationship is constrained by the connection state, and the constrained relationship can be described by a non-linear function. Suppose the length of the flexible connection cable is l m, and the measured values of the GPS and depth sensor are ${\left[fx_{ik},fy_{ik},fz_{ik}\right]}^{T}\in R^{3}$. We construct a nonlinear equation between observations and state variables and the specific expression is as follows:(3)$$\left[\begin{array}{l} fx_{ik}\\ fy_{ik}\\ fz_{ik} \end{array}\right]=h\left(\left[\begin{array}{l} u_{ik}\\ {u^{\prime }}_{ik} \end{array}\right],l\right)+\delta_{k}$$
where $\delta_{k}$ is the measurement noise, $fz_{ik}$ is the measurement value of the depth sensor, $\left(fx_{ik},fy_{ik}\right)$ are the GPS measurements in the geodetic coordinate system, and h(.) represents the relationship between state variables and observation variables, the specific expression of which is shown as follows:(4)h([uiku′ik],l)=[uxik+l2-uzik2*sinθuyik+l2-uzik2*cosθ        uzik]=[uxik+l2-uzik2*sin(arctanuvixkuviyk)uyik+l2-uzik2*cos(arctanuvixkuviyk)        uzik]

Through Equations (1)~(4), taking the position and speed of the modem as state variables, and the GPS value of the buoy as observation variables, we construct a nonlinear state filter model and use UKF to estimate the high-precision position of the buoy modem in real-time. Then, the beacon modem sends its position to the underwater mobile node through the location signal.

### 3.2. Arrival Time Calibration of the Location Signal

The mobile node collects location signals during the positioning process and analyzes the arrival time through matched filtering. As shown in Figure 3, because the propagation time of the location signal is different, the arrival time and position of the mobile node receiving different location signals have changed. This means the mobile node cannot collect all positioning signals at the same time or at the same position.

This paper uses time-delay calculation to calibrate the location signal arrival time. The main idea is that the measurement delay is not directly observed by the received location signals but obtained by compensating the observation results to the same receiving time or the same location [20]. Specifically, the calculation process is to firstly establish a linear relationship between the mobile node position and the measurement delay (containing clock bias of the asynchronous system), and then use this relationship to interpolate to obtain the compensated measurement time.

Under the asynchronous system shown in Figure 1, considering the ith beacon and the jth localization cycle, the ith beacon sends a location signal at the instant jτ, where τ is the cycle period of the system, and the mobile node receives the signal at the receiving instant $T_{ij}=\left(j\tau +t_{ij}\right)\times \delta_{a}+\delta_{b}$, where $\delta_{a}$ and $\delta_{b}$ are the clock skew and offset, respectively, of the mobile node with respect to beacons and $t_{ij}$ is the real propagation delay. We define $tp_{ij}=t_{ij}\times \delta_{a}+\delta_{b}$ as the measurement delay at the mobile node position corresponding to the receiving instant $T_{ij}$. We use the measurement delay values of two consecutive cycles to construct a linear relationship. As shown in Figure 4, the eight solid points ($tp_{i\left(j-1\right)}\;and\;tp_{ij},i\in [1,4]$) correspond to the measured values of two consecutive positioning cycles (j−1~j). The four curves correspond to four linear relationships established by the mobile node position and the measurement delay. According to these relations, we can interpolate the measurement delay of the mobile node at any position.

It is assumed that the propagation delay between the beacon 1 and the mobile node is small, namely $t_{1j}$ is the smallest of $t_{ij}$. The mobile node needs to compensate $tp_{ij}$ to the mobile position corresponding to the reference instant $T_{1j}=j\tau +t_{1j}+\delta$. The measurement delay $t{\bar{p}}_{ij}$ after the interpolation process is as follows:(5)$$t{\bar{p}}_{ij}=tp_{ij}-\frac{T_{ij}-T_{1j}}{T_{ij}-T_{i(j-1)}}*(tp_{ij}-tp_{i(j-1)}),\;i\in \left[1,4\right]$$

Using the compensated measurement delay $t{\bar{p}}_{ij}$, the mobile node can obtain the high-precision position information of the mobile node at the reference time $T_{1j}$.

### 3.3. High-Precision Position for Mobile Node

The motion state of the underwater mobile node is relatively simple, and mostly the motion is uniform linear motion, so a constant speed model is used for modeling the motion of the mobile node. The position ${\left[x_{k}\;,y_{k}\right]}^{T}$ and velocity ${\left[v_{xk},\;v_{yk}\right]}^{T}$ of the mobile node are used to represent state variables, and the state equation is:(6)$$\left[\begin{array}{l} x_{k}\\ y_{k}\\ v_{xk}\\ v_{yk} \end{array}\right]=\left[\begin{array}{l} 1\;\;0\;\;T\;\;0\\ 0\;\;1\;\;0\;\;T\\ 0\;\;0\;\;1\;\;0\\ 0\;\;0\;\;0\;\;1 \end{array}\right]\left[\begin{array}{l} x_{k-1}\\ y_{k-1}\\ v_{xk-1}\\ v_{yk-1} \end{array}\right]+\left[\begin{array}{cc} T^{2}/2 & 0\\ 0 & T^{2}/2\\ T & 0\\ 0 & T \end{array}\right]\left[\begin{array}{l} a_{xk}\\ a_{yk} \end{array}\right]$$
where T is the update interval of the state variable; ${\left[a_{xk},\;a_{yk}\right]}^{T}$ are the random noise of the acceleration in the x and y direction, respectively; and its covariance matrix is Q. The measurement delay differences tc between different beacons are selected for observation variables. When calculating tc, $\delta_{b}$ can be eliminated directly, but the influence of $\delta_{a}$ still exists, for example, $tc_{21}=tp_{2j}-tp_{1j}=\left(t_{2j}-t_{1j}\right)*\delta_{a}$. The value of $\delta_{a}$ is very close to 1 and the deviation of the clock device we actually use between $\delta_{a}$ and 1 is no more than $2.5\times 10^{-5}$. Since the maximum value of $t_{mj}-t_{nj},\;\;m,n\in \left[1,\ldots ,4\right]$ is less than 7 s, the observation error of $tc_{mn}$ introduced by $\delta_{a}$ is very small, and the maximum error is no more than 0.175 ms, which has little impact on the positioning results. Therefore, in the later analysis of this paper, $\delta_{a}$ is approximately 1. The measurement equation is constructed through ray propagation theory. Due to the sound ray bending phenomenon caused by the complex ocean environment, the measurement equation is a nonlinear equation, which is shown as follows:(7)$$tc=H\left(S_{k}\right)+w_{k}$$

A common method for constructing measurement equations in underwater target positioning systems is to approximate the sound rays as straight lines and replace the sound velocity with the average sound velocity c. Thus, the measurement equation can be expressed as:(8)$$\left[\begin{array}{l} tc_{21}\\ tc_{31}\\ tc_{41} \end{array}\right]=\left[\begin{array}{l} tp_{2k}-tp_{1k}\\ tp_{3k}-tp_{1k}\\ tp_{4k}-tp_{1k} \end{array}\right]=\left[\begin{array}{l} \left\|S_{k}-\left(x_{1},y_{1},z_{1}\right)\right\|-\left\|S_{k}-\left(x_{2},y_{2},z_{2}\right)\right\|/c\\ \left\|S_{k}-\left(x_{1},y_{1},z_{1}\right)\right\|-\left\|S_{k}-\left(x_{3},y_{3},z_{3}\right)\right\|/c\\ \left\|S_{k}-\left(x_{1},y_{1},z_{1}\right)\right\|-\left\|S_{k}-\left(x_{4},y_{4},z_{4}\right)\right\|/c \end{array}\right]+w_{k}$$
where $S_{k}=\left(x_{k},y_{k},z_{k}\right)$, $z_{k}$ can be obtained by the depth sensor; $w_{k}$ is the time measurement error and its covariance matrix is R; and ‖.‖ represents the distance between two points. This method does not consider the sound ray bending problem yet introduces a sound velocity error and increases the positioning error.

Regarding sound ray bending, the conventional methods are the sound ray correction method and the effective sound velocity method. The sound ray correction method is an iterative method based on ray acoustics. However, the calculation process of this method is complicated, especially under complex hydrological conditions and when the sound ray undergoes multiple sea surface reflections; thus, it cannot be applied to signal processing boards with limited calculation capabilities for mobile nodes. The effective sound velocity method [21] is to use the sound velocity profile information obtained in advance to calculate the distance-depth-equivalent sound–velocity relationship between the transmitting and receiving nodes, and then store this relationship in the underwater mobile node to correct the sound ray bending problem before deployment.

Based on the idea of effective sound velocity, this paper proposes a method for constructing measurement equations considering the bending of sound rays. Using environmental information and the BELLHOP model, the distance-depth-delay relationship between the underwater platform and the beacon is established in advance. Through this relationship, the measurement equation between state variables and observation variables is established as follows:(9)$$\left[\begin{array}{l} tc_{21}\\ tc_{31}\\ tc_{41} \end{array}\right]\;=\;\left[\begin{array}{l} t{\bar{p}}_{2k}-t{\bar{p}}_{1k}\\ t{\bar{p}}_{3k}-t{\bar{p}}_{1k}\\ t{\bar{p}}_{4k}-t{\bar{p}}_{1k} \end{array}\right]=t_{e}\left(z,z_{1},\left\|S_{k}-\left(x_{1},y_{1},z_{1}\right)\right\|\right)-t_{e}\left(z,z_{i},\left\|S_{k}-\left(x_{i},y_{i},z_{i}\right)\right\|\right)+w_{k}$$
where $t{\bar{p}}_{ij}$ is the compensated measurement delay; $t_{e}\left(z_{i},z,r\right)$ represents the distance-depth-delay relationship; and z, $z_{i}$, and r represent the depth of the beacon, the depth of the underwater target, and the range, respectively. The method of establishing this relationship is to first determine the range of z, $z_{i}$, and r, and its corresponding step size Δz, $\Delta z_{i}$, and Δr according to the actual situation. Then, the environmental parameters and the parameters of the beacon and underwater target are input into the BELLHOP model, and thirdly, the earliest arrival time of the signal in each case is extracted. Finally, a functional relationship is established.

## 4. Posterior Cramér–Rao Bound (PCRB)

The lower bound of the mean squared error for any discrete-time filtering can be computed by PCRB [22]. The author in [23] provides a formula for updating the posterior Fisher information matrix (FIM) from one time instant to the next. The posterior FIM sequence for a linear process and a non-linear measurement model can be computed as:(10)$$\begin{array}{cc} J_{k} & ={\left(Q_{k}+FJ_{k-1}^{-1}F^{T}\right)}^{-1}+E\left\{\nabla_{x_{k+1}}H_{k+1}^{T}R_{k+1}^{-1}\nabla_{x_{k+1}}^{T}H_{k+1}^{}\right\}\\ & ={\left(Q_{k}+FJ_{k-1}^{-1}F^{T}\right)}^{-1}+H_{k}^{T}R_{k}^{-1}\nabla_{x_{k+1}}^{T}H_{k}^{} \end{array}$$
where $Q_{k}$ is the state noise covariance matrix, $R_{k}^{}$ is the measurement error covariance matrix, and $H_{k}^{}$ is the measurement Jacobian matrix. Since we basically estimate the location of the mobile node, namely the first two items in the state variable, the PCRB of the final estimates will correspond to the sum of the first two diagonal elements as:(11)$$PCRB_{k}=\sum_{i=1}^{2}{\left[J_{k}^{-1}\right]}_{ii}$$

## 5. Experiments and Analysis

The following simulation and sea trial results are carried out to investigate the localization accuracy of the proposed algorithm.

### 5.1. Simulation Experiments and Analysis

Four floating buoy nodes (buoy 1 to buoy 4) are used to locate an AUV in real-time in a simulation experiment. The floating trajectory of the buoy and the motion trajectory of the AUV are shown in Figure 5. The specific simulation conditions are listed as follows:(1)The four buoy nodes form a square topology with a side length of 6 km, and the horizontal initial positions of the four buoys are (3000, −3000 m), (−3000, −3000 m), (3000, 3000 m), (−3000, 3000 m).(2)Each buoy node is equipped with GPS equipment. The GPS equipment can obtain the horizontal position of the buoy body in real-time, and there is a Gaussian distribution error in the position information with a mean of 0 m and a variance of 1 m^2^.(3)The buoy body drifts with the current. The drifting directions of the four buoys are the same, but the velocities are different, and they all obey a Gaussian distribution.(4)Each buoy node is equipped with an underwater acoustic modem. The modem and the buoy body are softly connected by a cable, and the length is 10 m. The modem depth is 8 m. The drifting trajectory of the modem is the same as that of the buoy body, and the measured value of the depth sensor obeys Gaussian distribution N (8 m, 0.1 m^2^).(5)The movement trajectory of the AUV is a parabola. For the trajectory, its initial horizontal position is (−3000, 2675 m), the horizontal position of the inflection point is (510, 1000 m), and the horizontal position of the endpoint of movement is (3000, −875 m). The depth of the underwater mobile node obeys a Gaussian distribution N (50 m, 0.2 m^2^).(6)The real propagation time of the positioning signal received by the underwater mobile node is obtained through the ray propagation model, and a Gaussian distribution error is added to the time measurement result.(7)The water depth of the simulation area is 100 m and medium hydrological conditions are selected.(8)The simulation lasts 6000 s, and the positioning period is 10 s; thus, there are 600 positioning cycles in the simulation.

Figure 6 is the result of the modem position of buoy 1. The blue line with * is the GPS direct measurement result, the red line with o is the real position of the modem of buoy 1, and the yellow belt + line corresponds to the result after the space position calibration of Section 3.1 is performed. It can be seen from Figure 6 that the position of the modem obtained after the spatial position calibration is close to the true position while the GPS direct measurement result has a large deviation from the real position of the modem. The corresponding root mean square (RMS) values at different coordinates are used to evaluate the performance of the GPS value and the calibrated value:(12)$$\begin{array}{l} RMS\_x=\sqrt{\frac{1}{N}\sum_{k=1}^{N}{\left(x_{k}-{\hat{x}}_{k}\right)}^{2}}\\ RMS\_y=\sqrt{\frac{1}{N}\sum_{k=1}^{N}{\left(y_{k}-{\hat{y}}_{k}\right)}^{2}}\\ RMS\_r=\sqrt{\frac{1}{N}\sum_{k=1}^{N}{\left(x_{k}-{\hat{x}}_{k}\right)}^{2}+{\left(y_{k}-{\hat{y}}_{k}\right)}^{2}} \end{array}$$
where $\left(x_{k},y_{k}\right)$ is the estimated horizontal position, and $\left({\hat{x}}_{k},{\hat{y}}_{k}\right)$ is the true horizontal position of the modem.

The RMS results are analyzed in Table 1 and Figure 7. The modem position after spatial position calibration has much smaller average and maximum values when compared with the GPS measurement. It can be seen from Figure 7 that the position error of the modem is significantly reduced after spatial position calibration.

The time delay calibration results of the positioning signal are shown from Figure 8, Figure 9 and Figure 10. Figure 8 is the relative measurement delay result of the underwater mobile node receiving the positioning signal of each buoy in the whole simulation process. Figure 9 is an enlarged result of Figure 8 around 1000 s, and the four points corresponding to the dotted line are the relative measurement delay obtained after using the motion compensation algorithm in Section 3.2. The measurement delay of the positioning signals is compensated by the corresponding position of AUV at 1003 s.

Using the ray model, we can calculate the true measurement delay of the positioning signals when the AUV is in any position. Figure 10 shows the comparison result of the measured delay error with and without the time delay calibration. It can be seen that after time calibration, the measurement delay error is significantly smaller.

Figure 11 shows the positioning results of the AUV with different methods. The blue line in the figure is the positioning error obtained without the spatial-temporal calibration or the sound ray correction. With sound ray correction, the orange + line and the red O line correspond to the positioning result after spatial calibration and time calibration, respectively. The purple ∗ line is the positioning error corresponding to the algorithm proposed in this article, and the blue ⊲ line is the positioning error corresponding to the PCRB threshold. Table 2 shows the statistical results of the horizontal error. Methods 1 to 4 in the table correspond to the methods represented by the first four curves in the annotation box in the upper right corner of Figure 11. It can be seen from the data in the table that the algorithm (method 4) proposed in this paper reduces the positioning error and improves the horizontal positioning accuracy.

It can be seen that with the use of the sound ray correction, spatial calibration, and time calibration algorithms, the positioning error of the mobile node can be reduced. After sound ray correction and spatial-temporal self-calibration, the proposed underwater acoustic network positioning method can achieve a positioning error that is close to the PCRB error. The simulation results verify the effectiveness of the algorithm proposed in this paper.

### 5.2. Sea Trial Analysis

To verify the effectiveness of the method proposed in this paper, a sea trial verification experiment was carried out in the East China Sea. During the sea trial, four buoy nodes and an AUV node were used. Each buoy node contained a modem and an anchor block. The diameter of the buoy was 1.2 m and the weight was 500 kg. The weight of the anchor block was 100 kg. The AUV node was simulated by a surface ship, and the surface ship was equipped with differential GPS (DGPS). The DGPS output value was used as the real position of the AUV and was compared with the positioning result of the algorithm proposed in this paper to analyze the positioning performance. The movement trajectory of the AUV and the placement position of the buoy beacons are shown in Figure 12. The maximum distance between the buoys was about 10.2 km. All four buoys drifted with waves, and the drifting trajectory is shown in Figure 13. The drifting trajectories of the four buoys are similar.

Figure 14 is the result of the modem position of the buoy 3. The blue line with ∗ in the figure is the GPS measurement result, and the red line with O is the position of the buoy modem after spatial position calibration. It can be seen that the compensated modem position lags a certain distance behind the buoy drifting trajectory. Figure 15 shows the results of the drifting speed of the buoy 3. Due to the influence of the anchor block, the drifting speed is relatively low.

The result of the positioning signal after delay calibration is shown in Figure 16. The dotted line in the figure corresponds to the measurement delay of the 70th positioning cycle. The enlarged view in the lower right corner corresponds to the measurement delay after compensation.

Figure 17 and Figure 18 and Table 3 show the positioning results and error analysis of the sea trial. In Figure 17, the solid green line is the AUV trajectory collected by DGPS. The corresponding methods for the other colored lines are the same as in Figure 11. Figure 18 is the horizontal error of different positioning cycles, and the calculation method is $err\_r_{k}=\sqrt{{\left(x_{k}-{\hat{x}}_{k}\right)}^{2}+{\left(y_{k}-{\hat{y}}_{k}\right)}^{2}}$, where $\left(x_{k},y_{k}\right)$ is the estimated horizontal position, and $\left({\hat{x}}_{k},{\hat{y}}_{k}\right)$ is the true horizontal position of the modem. Table 3 shows the statistical results of the horizontal error. Methods 1 to 4 in the table correspond to the methods represented by the first four curves in the annotation box in the upper right corner of Figure 17. It can be seen from the data in the table that the algorithm (method 4) proposed in this paper reduces the positioning error and improves the horizontal positioning accuracy.

It can be seen that with the sound ray correction, spatial calibration, and time calibration algorithm, the positioning result of the mobile node is closer to the true trajectory, and the positioning error is less. The underwater acoustic network positioning method based on spatial-temporal self-calibration proposed in this paper can effectively reduce the positioning error of underwater mobile nodes and improve the positioning accuracy after sound ray correction and spatial-temporal calibration.

## 6. Conclusions

In the process of networked positioning, the buoy modem will drift with waves, and the movement of the underwater node will affect the positioning results and increase positioning error. This paper proposes an underwater acoustic network localization method based on spatial-temporal self-calibration. The following conclusions can be drawn from our analysis and comparison:(1)The real-time position of the buoy modem is affected by current and is difficult to accurately obtain. To solve this problem, this study presented a real-time compensation method for the buoy modem position. In the presence of the modem position offset with the flow, the compensation method can accurately estimate the modem space position through the soft connection relationship between the buoy modem and the buoy body.(2)The movement of the underwater node can increase the time delay error, so this paper proposes a time delay calculation method. The main idea is to normalize the ranging information to the same sampling time, which can reduce the measurement delay error.(3)Under the influence of sound ray bending, the positioning error of the underwater mobile node is large. To solve this problem, a networked positioning model based on the effective sound velocity was proposed. No matter how complex the sea environment is, the positioning model can revise the sound ray in real-time and achieve the high-precision positioning of mobile nodes. Both the simulation results and experimental data verify the effectiveness of the algorithm proposed in this paper.

## Figures and Tables

**Figure 1 sensors-22-05571-f001:**
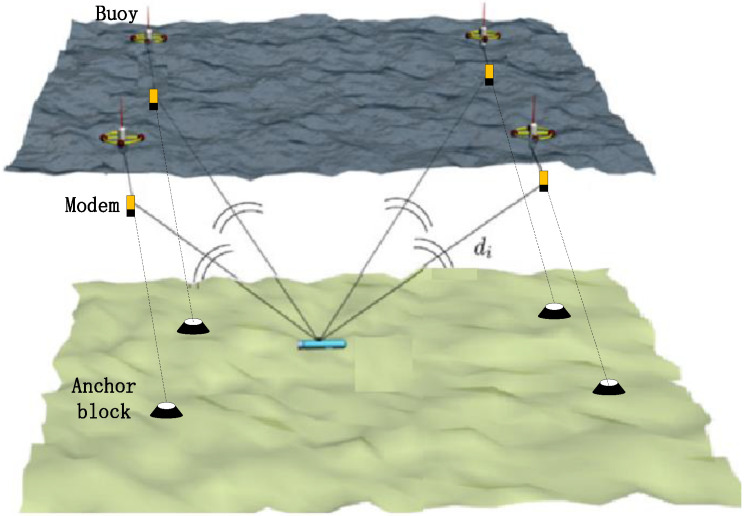
Structure of the positioning system schemes.

**Figure 2 sensors-22-05571-f002:**
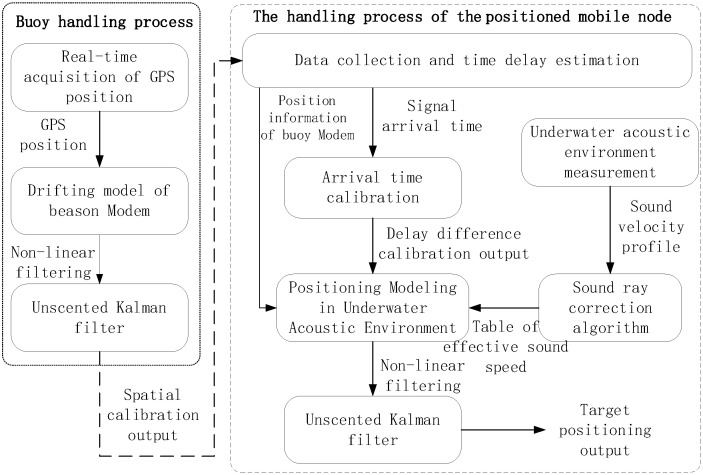
Working principle of the positioning system.

**Figure 3 sensors-22-05571-f003:**
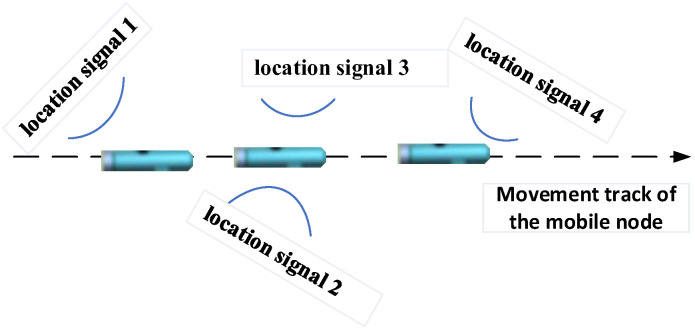
The schematic diagram of asynchronous sampling of the mobile node detection location signals.

**Figure 4 sensors-22-05571-f004:**
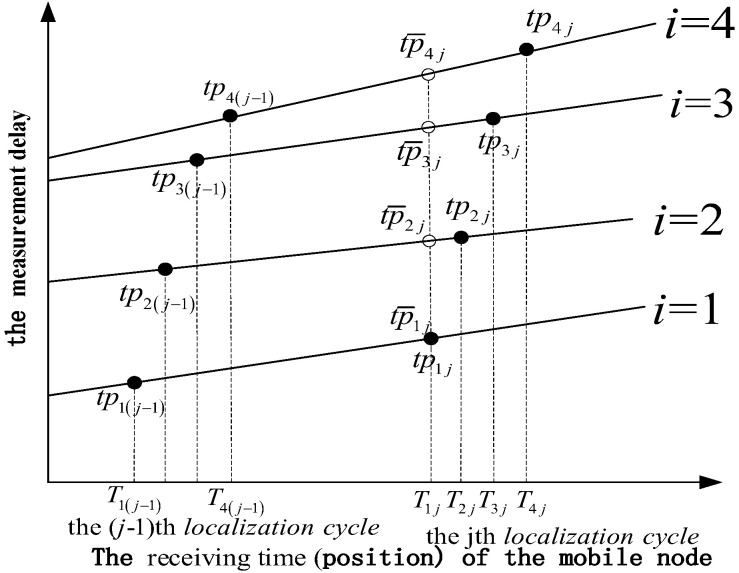
The schematic diagram of the time delay calibration method.

**Figure 5 sensors-22-05571-f005:**
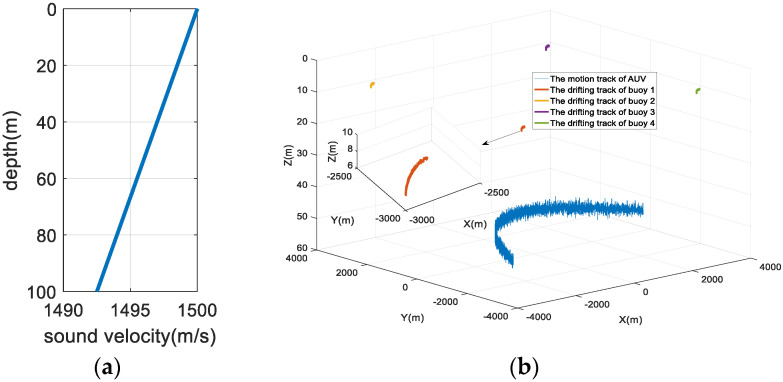
Simulation conditions. (**a**) The simulated sound velocity profile; (**b**) the floating trajectory of the buoy and the motion trajectory of the underwater mobile node.

**Figure 6 sensors-22-05571-f006:**
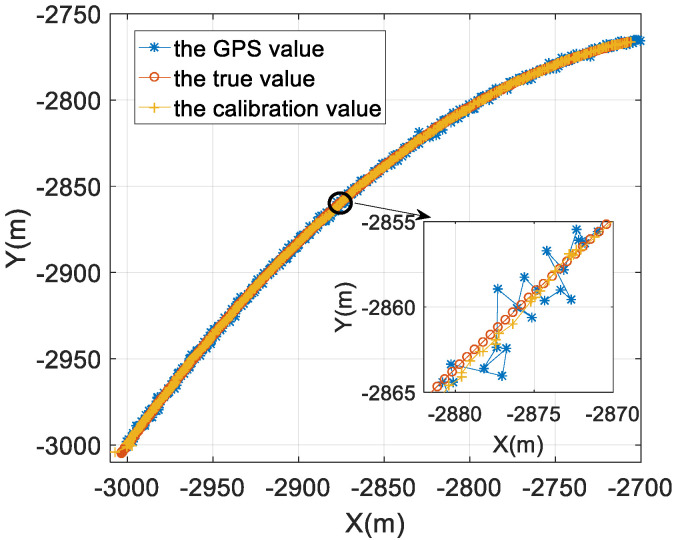
The analysis result of the modem position of buoy 1.

**Figure 7 sensors-22-05571-f007:**
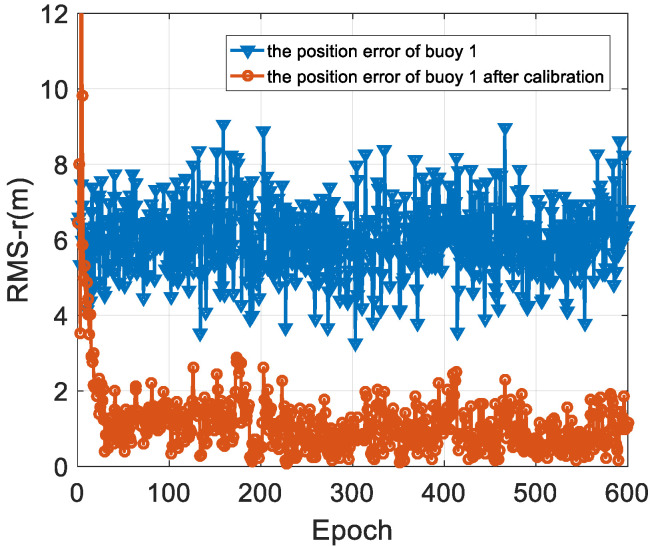
The positioning error analysis result of buoy 1.

**Figure 8 sensors-22-05571-f008:**
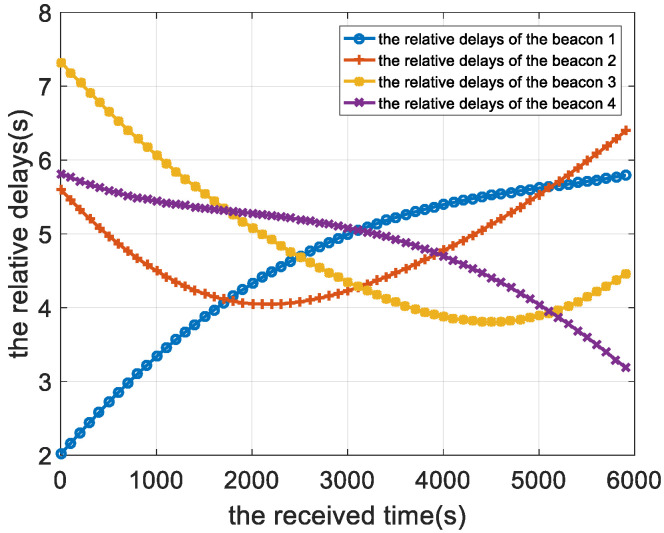
Relative time delay result of the location signal.

**Figure 9 sensors-22-05571-f009:**
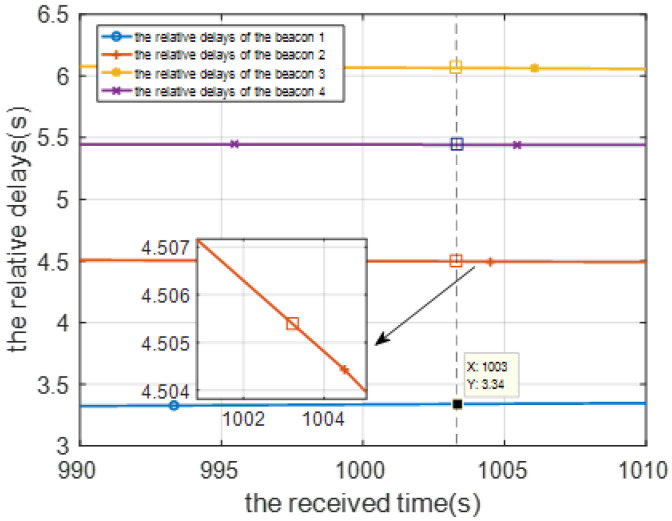
Relative propagation delay after motion compensation.

**Figure 10 sensors-22-05571-f010:**
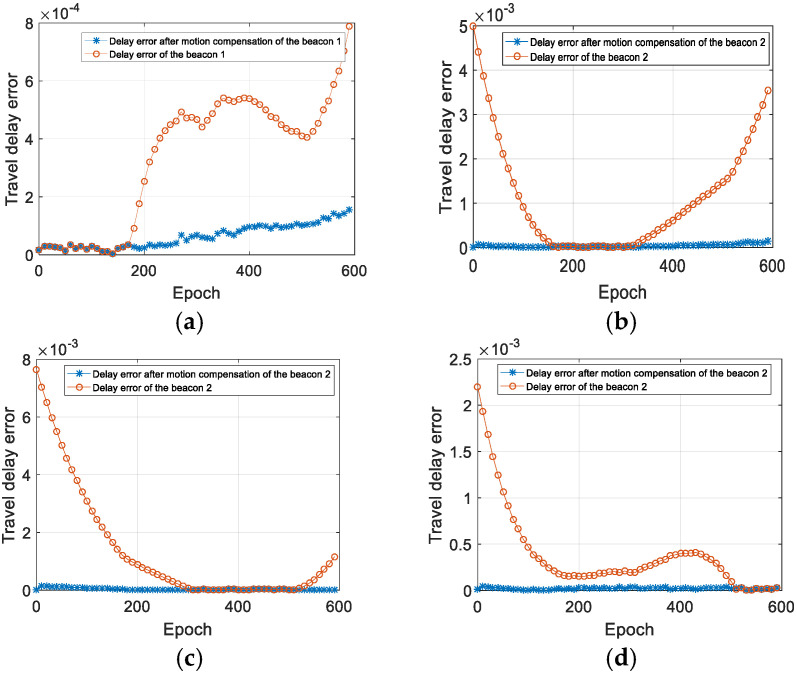
Analysis of the measurement delay error. (**a**) The delay error of the buoy 1; (**b**) the delay error of the buoy 2; (**c**) the delay error of the buoy 3; (**d**) the delay error of the buoy 4.

**Figure 11 sensors-22-05571-f011:**
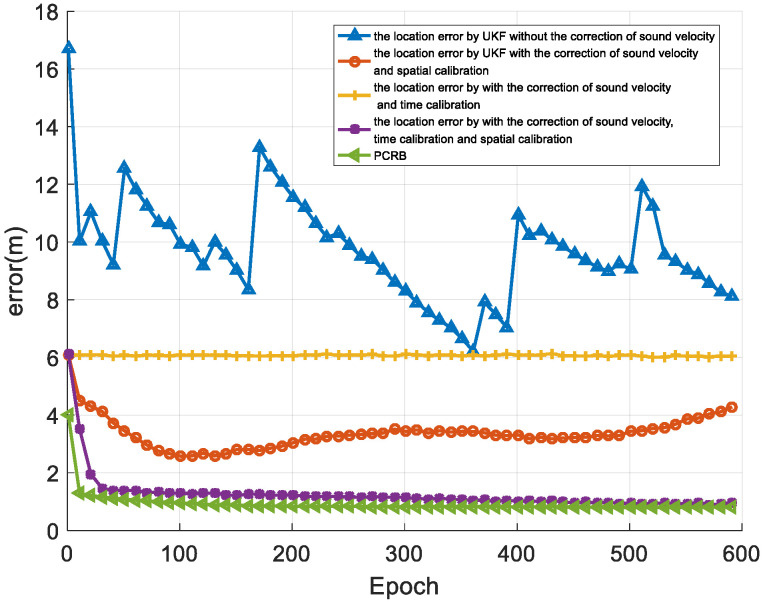
The positioning results of different methods.

**Figure 12 sensors-22-05571-f012:**
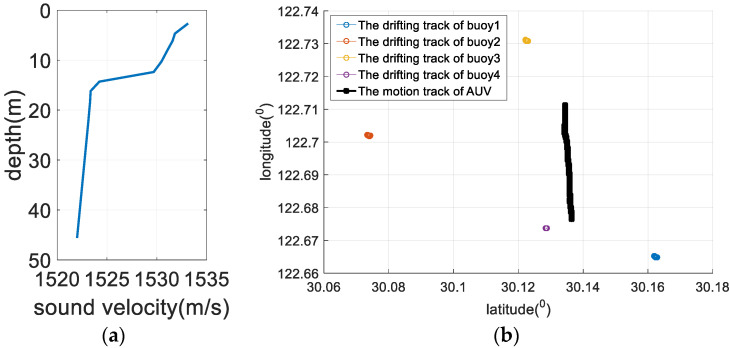
Sea trial conditions. (**a**) The sea trial sound velocity profile; (**b**) the movement trajectory of the AUV and the position of the buoy beacons.

**Figure 13 sensors-22-05571-f013:**
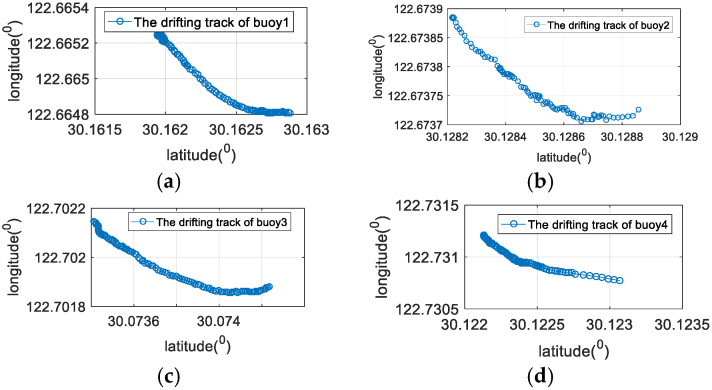
The drifting track of four buoy beacons. (**a**) The drifting track of the buoy 1; (**b**) the drifting track of the buoy 2; (**c**) the drifting track of the buoy 3; (**d**) the drifting track of the buoy 4.

**Figure 14 sensors-22-05571-f014:**
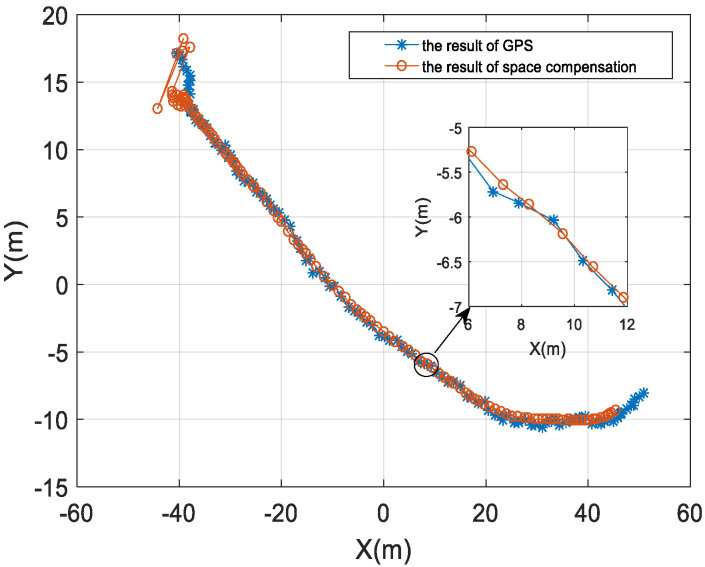
The analysis result of the modem position.

**Figure 15 sensors-22-05571-f015:**
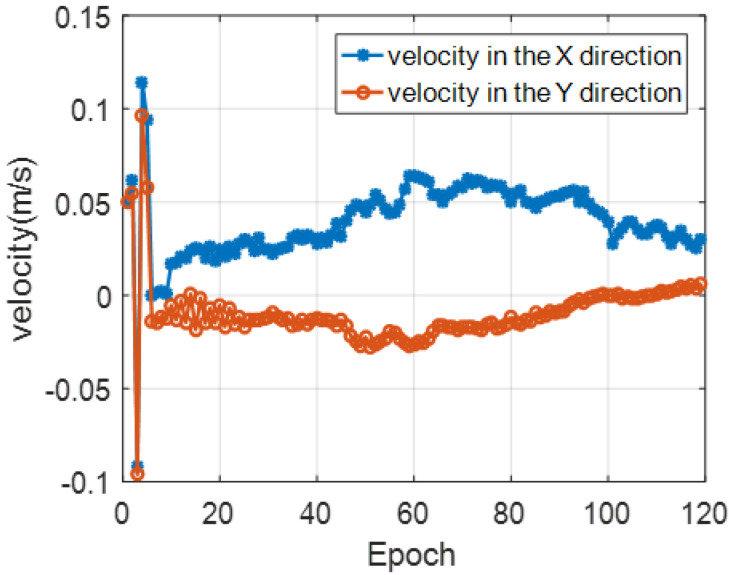
The analysis result of the modem drifting speed.

**Figure 16 sensors-22-05571-f016:**
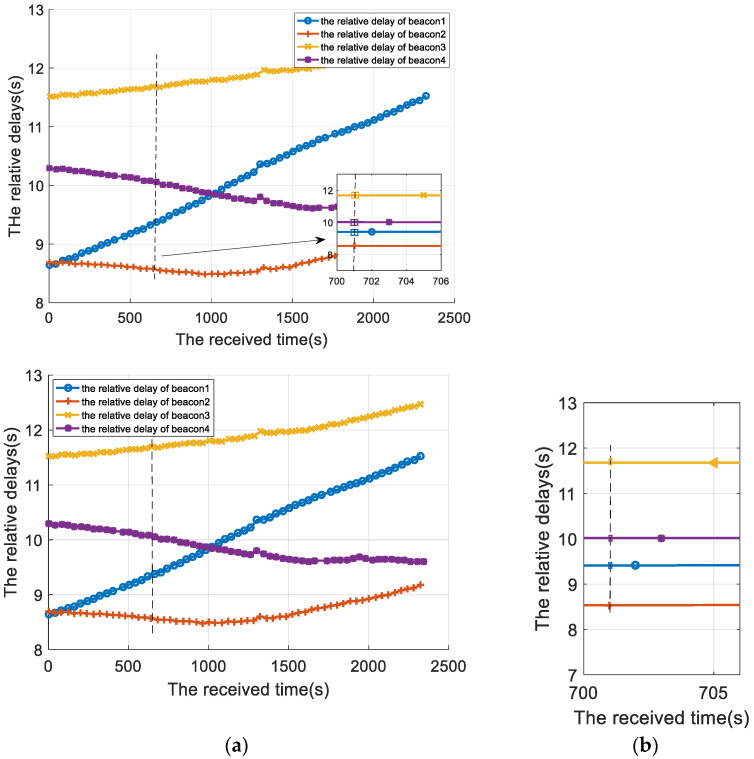
The relative delay result of the positioning signal after delay calibration (**a**) The relative delay of four positioning signal after delay calibration; (**b**) the relative delay of the 70th positioning cycle.

**Figure 17 sensors-22-05571-f017:**
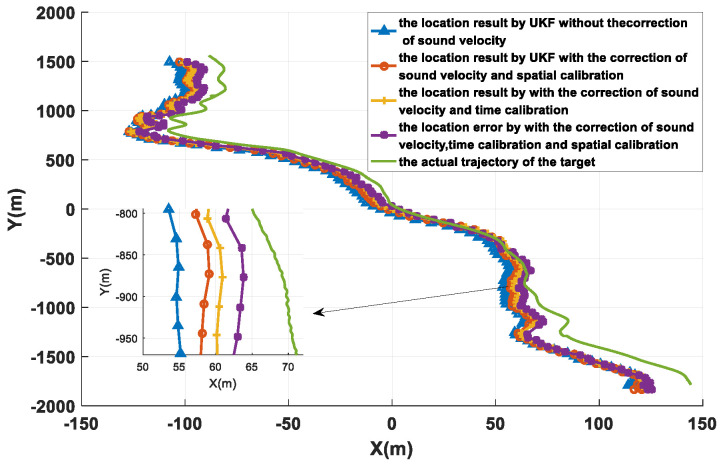
The positioning results of different methods.

**Figure 18 sensors-22-05571-f018:**
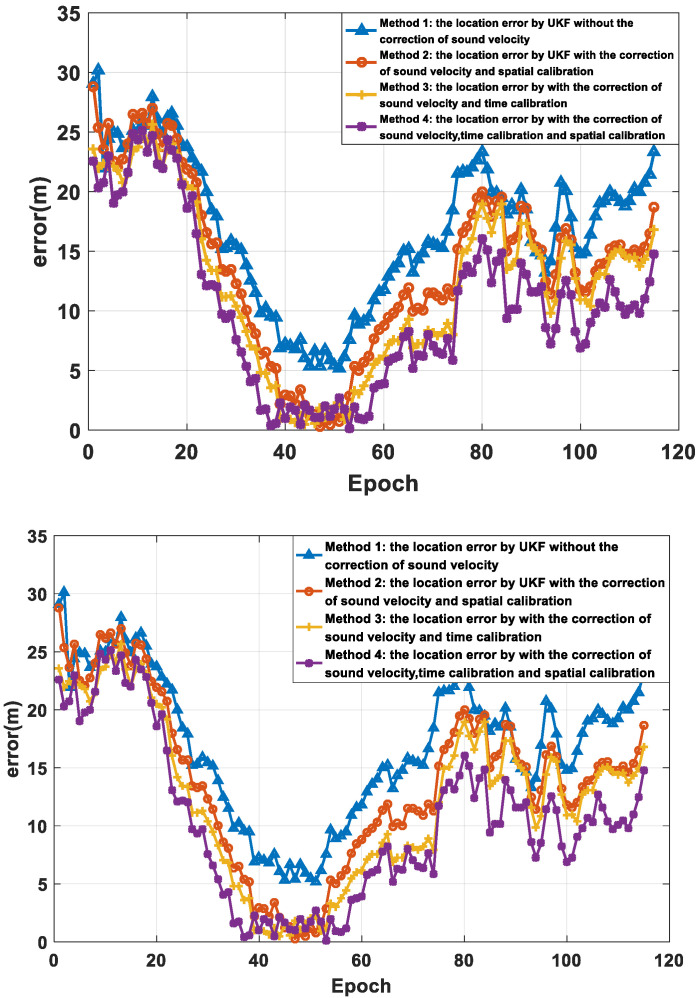
The error analysis of different methods.

**Table 1 sensors-22-05571-t001:** The RMS analysis results of buoy 1.

Statistics	Method	RMS_x (m)	RMS_y (m)	RMS_r (m)
Min	GPS value	4.44	3.78	6.03
	Calibration value	0.81	0.75	1.15
Max	GPS value	8.51	8.02	9.06
	Calibration value	2.02	2.13	2.90

**Table 2 sensors-22-05571-t002:** Comparison of the position results of different methods.

Method	Mean (m)	Std (m)	Max (m)	Min (m)
Method 1	9.59	1.53	13.27	6.21
Method 2	3.30	0.41	4.31	2.57
Method 3	6.06	0.03	6.13	6.01
Method 4	1.13	0.18	1.95	0.88

**Table 3 sensors-22-05571-t003:** Comparison of the position results of different methods.

Method	Mean (m)	Std (m)	Max (m)	Min (m)
Method 1	16.86	6.22	30.13	5.2
Method 2	13.81	7.27	28.8	0.26
Method 3	12.2	7.06	25.64	0.48
Method 4	10.33	6.99	25.14	0.11

## Data Availability

Not applicable.

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
