# Peer review of "An Underwater Acoustic Network Positioning Method Based on Spatial-Temporal Self-Calibration"

_sensors, 2022, doi:10.3390/s22155571_

Round 1

Reviewer 1 Report

In this manuscript: An underwater acoustic network positioning method based on 2

spatial-temporal self-calibration, authors include very fewe recent and realted Works, only two of them are from the last three years:

2. Su, X.; Ullah, I.; Liu, X.; Choi, D. A Review of Underwater Localization Techniques, Algorithms, and Challenges. Journal of 458
Sensors, 2020, 4, 1-24.

11. Ullah, I.; Qian, S.; Deng, Z.; Lee, J. Extended Kalman Filter-based Localization Algorithm by Edge Computing in Wireless Sensor 475
Networks. Digital Communications & Networks (DCAN) , 2021, 7, 187-195.

There any references of this year, and the number of total references is so reduced for a hot topic. You are encouraged to revise recent and related Works to justify the state of the art in this problem being solved to compare your proposed solution.

You are emphasizing the use of Kalman filter in the sentence in your abstract: Under the asynchronous system, the 17 influence of the inhomogeneity of the underwater medium is analyzed, and the unscented Kalman 18 algorithm is used to estimate the position of underwater mobile nodes. Finally, the effectiveness of 19 this method is verified by simulation and sea trials… Can you discuss the comparison with other kalman filters, as the ones already applied in the reference: Kalman observers in estimating the states of chaotic neurons for image encryption under MQTT for IoT protocol, The European Physical Journal Special Topics 231 (5), 945-962, 3, 2022

In yoour introduction you discuss posible target values to improve your work. They are given in the paragraph: People have developed many different positioning methods [1-5], taking advantage 39 of the feature that acoustic signals can travel long distances in the underwater environ- 40 ment. The main idea of these methods is to obtain distance information from the reference 41 node according to sound propagation characteristics, and then convert the distance infor- 42 mation into the underwater node’s position information through signal processing. Dis- 43 tance information is usually characterized by signal strength (SS), angle of arrival (AOA), 44 time of arrival (TOA), and time differences of arrival (TDOA) [1,2]. In the underwater 45 acoustic environment, the strength of the received signal is not convenient since the prop- 46

agation loss is difficult to obtain accurately in a time-varying environment [5]. Using AOA 47 for UANS has been considered in [6], but it has not been widely employed due to the size 48 and cost of directional antennas. Distance measurement based on propagation time delay 49 is widely used in underwater acoustic networks. This is because the speed of sound prop- 50 agation in water is low (about 1500m/s), and there is no need to receive signals with high 51 time resolution. This paper is also based on the differences in signal arrival time for un- 52 derwater acoustic network localization… Can you provide a list of specifications in order to see if your work is impriving them? But you should include a comparison with very recent and related Works if posible.

The experiment decribed in the paragraph: 5.2. Sea Trial Analysis 367 To verify the effectiveness of the method proposed in this paper, a sea trial verifica- 368 tion experiment was carried out in the East China Sea. During the sea trial, four buoy 369 nodes and an AUV node are used. Each buoy node contains a modem and an anchor block. 370 The diameter of the buoy is 1.2m and the weight is 500kg. The weight of the anchor block 371 is 100kg. The AUV node was simulated by a surface ship, and the surface ship was 372 equipped with differential GPS (DGPS). The DGPS output value was used as the real po- 373 sition of the AUV and was compared with the positioning result of the algorithm pro- 374 posed in this paper to analyze the positioning performance. The movement trajectory of 375 the AUV and the placement position of the buoy beacons are shown in Figure 12. The 376 maximum distance between the buoys was about 10.2km… these details are interesting and should be compared with similar Works in a Table. Also your simulations provided in all your figures can be summarized in a Table, this will highlight the three contributions listed in your conclusions.

Author Response

Those comments are all valuable and very helpful for revising and improving our paper, as well as the important guiding significance to our research. We have studied the comments carefully and have made corrections which we hope meet with approval. See the attached word for the specific reply. Thank you very much.

Reviewer 2 Report

1.The authors propose an underwater acoustic network localization (positioning) method based on spatial-temporal self-calibration. In the method the space position of the beacon modem is automatically calibrated using only the GPS position and depth sensor information obtained in real-time.

2.The examined real-life problem is dynamical  (i.e., the movement of the underwater nodes that leads to increasing  the time delay errors). The authors main idea is based on normalization of the ranging information to the same sampling time, which can reduce the measurement delay error.  

3.The unscented Kalman algorithm is used to estimate the position of underwater mobile nodes.

4.The effectiveness of the suggested method is verified: (i) by  simulation experiments and analysis, and (ii) by sea trial analysis.

5.Generally, the paper material is prepared at a very good level (all parts): (a) the description of the architecture of the Network Position System and principles of the system, (b) the description of the suggested method, and (c) the verification of the method (i.e., simulation and analysis). The paper can be accepted.

6.It may be interesting (in the future) to consider some optimization positioning models as well.

Author Response

(The authors gave the same response as above.)

Author Response

(The authors gave the same response as above.)
